# *Fusobacterium nucleatum* and Gastric Cancer: An Emerging Connection

**DOI:** 10.3390/ijms26167915

**Published:** 2025-08-16

**Authors:** Joana Sorino, Mario Della Mura, Giuseppe Ingravallo, Gerardo Cazzato, Cristina Pizzimenti, Valeria Zuccalà, Ludovica Pepe, Emanuela Germanà, Maurizio Martini, Antonio Ieni, Vincenzo Fiorentino

**Affiliations:** 1Section of Molecular Pathology, Department of Precision and Regenerative Medicine and Ionian Area (DiMePRe-J), University of Bari “Aldo Moro”, 70124 Bari, Italy; j.sorino@studenti.uniba.it (J.S.); mariodellamura1@gmail.com (M.D.M.); giuseppe.ingravallo@uniba.it (G.I.); gerycazzato@hotmail.it (G.C.); 2Anatomic Pathology Unit, Papardo Hospital, 98158 Messina, Italy; cristinapizzimenti86@gmail.com; 3Anatomic Pathology Unit, Department of Human Pathology in Adult and Developmental Age “Gaetano Barresi”, University of Messina, 98125 Messina, Italy; valeria.zuccala@unime.it (V.Z.); ludopepe97@gmail.com (L.P.); maurizio.martini@unime.it (M.M.); antonio.ieni@unime.it (A.I.); 4PhD Program in Translational Molecular Medicine and Surgery, Department of Biomedical, Dental, Morphological and Functional Imaging Sciences, University of Messina, 98125 Messina, Italy; emanuelagermana@hotmail.it

**Keywords:** *Fusobacterium nucleatum*, gastric cancer, microbiota, tumorigenesis, tumor microenvironment

## Abstract

*Fusobacterium nucleatum* (*F. nucleatum*), a Gram-negative anaerobe traditionally associated with periodontal disease, has recently emerged as a putative contributor to gastric carcinoma (GC) pathogenesis. Beyond its detection in gastric tissues, particularly in patients negative for *Helicobacter pylori* (*H. pylori*) or in advanced GC cases, *F. nucleatum* exerts diverse oncogenic effects. It promotes GC progression by modulating the tumor microenvironment through IL−17/NF-κB signaling, inducing tumor-associated neutrophils (TANs), upregulating PD-L1 expression, and enhancing immune evasion. Moreover, it increases tumor invasiveness via cytoskeletal reorganization, while extracellular vesicles (EVs) induced by the infection contribute to tumor cell proliferation, invasion, and migration. Clinically, its presence correlates with increased tumor mutational burden (TMB), venous thromboembolism, and poor prognosis. This review summarizes the current evidence regarding the emerging role of *F. nucleatum* in gastric tumorigenesis, examines its potential utility as a diagnostic and prognostic biomarker within the framework of precision oncology, and outlines the molecular methodologies presently employed for its detection in gastric tissue specimens.

## 1. Introduction

The human gastrointestinal tract harbors a dense and diverse microbial ecosystem, the gastrointestinal microbiota, consisting of more than 1500 microbial species—including bacteria, viruses, fungi, and protists [1,2]. This complex microbial network plays a critical role in maintaining host homeostasis by regulating metabolic pathways, supporting immune responses, and preserving epithelial barrier integrity [3,4,5]. The dominant bacterial phyla in the gastrointestinal tract include *Bacteroidetes* and *Firmicutes*, followed by *Proteobacteria*, *Fusobacteria*, *Tenericutes*, *Actinobacteria*, and *Verrucomicrobia* [6].

However, microbial equilibrium can be altered by external and lifestyle-related factors, such as diet, antibiotics, smoking, and physical inactivity, resulting in dysbiosis [7,8,9]. The latter is associated with a wide range of pathological conditions, such as metabolic disorders, inflammatory bowel diseases, and gastrointestinal malignancies, including GC [10,11].

Notably, GC remains a major global health challenge, representing the fifth most commonly diagnosed cancer and the third leading cause of cancer-related mortality worldwide [12]. Its incidence exhibits significant geographical variation, with the highest rates observed in Eastern Asia, Eastern Europe, and parts of Central and South America [12]. While the overall incidence has declined in recent decades, largely due to improved sanitation and treatment of *H. pylori* infection, GC’s prognosis remains poor, especially when diagnosed at advanced stages. This underscores the urgent need to understand all contributing factors to its pathogenesis, including the complex interplay between the host and the gastric microbiome. While *H. pylori* has long been recognized as the primary microbial initiator of GC [13,14], its presence is not always observed and its abundance tends to decrease in advanced tumor stages [15], suggesting that other microbial species may take over during tumor progression. In this regard, recent studies have highlighted the enrichment of non-*H. pylori* pathobionts—such as *Streptococcus anginosus*, *Prevotella*, and especially *F. nucleatum*—in GC tissues, in particular in *H. pylori*-negative cases [16,17]. In parallel, a progressive loss in diversity and richness across gastric, oral, and fecal microbiomes in GC patients has been observed [15].

Among the microbial species gaining increasing attention, *F. nucleatum*—a Gram-negative anaerobe and known contributor to colorectal cancer (CRC)—has emerged as a potential driver of gastric carcinogenesis and is believed to promote tumor progression through mechanisms such as immune modulation, immune evasion, proliferation, invasiveness, and migration of cancer cells [16,18,19,20].

This review critically examines the emerging role of *F. nucleatum* in gastric carcinogenesis and its clinical implications, and discusses its potential utility as a diagnostic and prognostic biomarker for GC patients. A focus on the current methodologies employed for its detection in gastric tissue specimens is also provided.

## 2. Materials and Methods

This study was carried out on public scientific databases (PubMed, Scopus, and Web of Science) up to May 2025, employing the following keywords: “*Fusobacterium nucleatum*,” “gastric cancer,” “gastric microbiota”, “gastric dysbiosis”, “gastric tumorigenesis”, “gastric tumor microenvironment”, and “*Helicobacter pylori*-independent gastric cancer”.

Only articles published in English were included. Studies were screened for relevance based on titles, abstracts, and full texts. Priority was given to original research articles with experimental, clinical, or metagenomic evidence on the presence or role of *F. nucleatum* in gastric tumorigenesis.

## 3. Biological and Pathogenic Features of *F. nucleatum*

*F. nucleatum* is a Gram-negative, anaerobic, non-spore-forming bacillus that is considered a prominent constituent of the oral microbiota. It plays a pivotal, context-dependent role in the formation and maintenance of oral biofilm, contributing to both periodontal health and disease. Within the complex architecture of dental biofilm, *F. nucleatum* serves as a structural intermediary—often referred to as a “bridge organism”—by facilitating interactions between Gram-positive bacteria (such as *Streptococcus* spp.) and Gram-negative, mostly anaerobic, colonizers such as *Porphyromonas gingivalis* (*P. gingivalis*). In fact, its characteristic elongated morphology supports simultaneous physical contact with multiple microbial partners, playing a central role in organizing polymicrobial communities [21,22]. In addition to its structural contributions, *F. nucleatum* interacts with both microbial species and host cells through a diverse repertoire of surface adhesins. Among these, RadD is the most well-characterized: it promotes co-aggregation with *Streptococcus mutans* by binding to the SpaP adhesin of the latter, enhancing biofilm maturation and complexity [23,24].

While commonly found in oral biofilm under healthy conditions, *F. nucleatum* also contributes to the development of periodontitis by modulating host immunity and enhancing pathogen virulence. In particular, it induces β-defensin 2 and pro-inflammatory cytokines such as IL−6 and IL−8 in oral epithelial cells [25,26,27], promoting inflammation that supports disease progression. As periodontitis is a polymicrobial disease, its interactions with other microbes are critical. Co-infection with *P. gingivalis* dampens inflammasome activation compared to *F. nucleatum* alone [28], and in turn, *F. nucleatum* enhances *P. gingivalis* invasiveness, suggesting synergistic mechanisms that favor immune evasion and chronic inflammation [21,29,30].

*F. nucleatum* exhibits not only adhesive and interactive capabilities but also invasive properties. Its major virulence factor, Fusobacterium adhesin A (FadA), binds to E-cadherin on epithelial cells and VE-cadherin on endothelial cells, promoting host cell invasion and potential systemic spread [31,32,33]. The translocation of *F. nucleatum* from the oral cavity to the gastrointestinal tract is a key event in its pathogenic mechanism, often described as the ‘oral–gut axis’ [34]. This journey can occur via two primary routes. The most direct is the gastrointestinal route, where bacteria are continuously swallowed with saliva, allowing them to reach the stomach and intestine [35,36]. While the stomach’s acidic environment is a formidable barrier, factors that raise gastric pH—such as *H. pylori*-induced atrophic gastritis or the use of proton pump inhibitors—can facilitate its survival and colonization [36]. Furthermore, recent studies have shown that *F. nucleatum* possesses intrinsic acid-resistance mechanisms, such as modifying its cell membrane composition with erucic acid, which enhances its survival in low-pH environments [18,36]. The second route is hematogenous, where bacteria enter the bloodstream through ulcerated oral tissues or during dental procedures, circulating systemically and seeding distant sites, including the tumor microenvironment [34,36,37].

Therefore, the diffusion of *F. nucleatum* can occur through direct dissemination as well as by transient bacteremia events, such as those induced by dental procedures [33]. To date, this microorganism has been isolated from various sites beyond the oral cavity in several pathological conditions, including preneoplastic and neoplastic diseases of the gastrointestinal tract [38,39,40].

The adhesin FadA binds to E-cadherin on the surface of epithelial cells. This interaction facilitates bacterial invasion and promotes tumor cell proliferation through activation of the Wnt/β-catenin signaling pathway. Moreover, the surface protein Fap2 interacts with TIGIT, an inhibitory receptor expressed on immune cells such as NK and T cells. This interaction suppresses their anti-tumor activity, enabling tumor cells to evade immune surveillance (Figure 1, created in BioRender. Ingravallo, G. (2025) https://BioRender.com/li3ppbl, accessed on 1 July 2025).

In this regard, the role of *F. nucleatum* involvement in carcinogenesis was first described in CRC, emerging as a relevant player. In fact, it was independently detected in CRC tissues via DNA and RNA sequencing [41,42], findings later confirmed by multiple molecular techniques including qPCR and FISH [41,43]. *F. nucleatum* localizes near colorectal crypts, occasionally intracellularly, and has been found to be viable in both primary tumors and matched liver metastases, suggesting that the microorganism may disseminate systemically alongside metastatic tumor cell migration [44,45]. Two surface bacterial proteins have been recognized as major actors involved in the adhesion process to epithelial cells as well as in the induction of tumorigenesis, thus representing both virulence and pathogenetic factors: FadA and Fap2 (Figure 1). FadA binds to E-cadherin in order to adhere to epithelial cells; in vitro experiments using CRC cell lines demonstrate the latter event in neoplastic cells induces β-catenin nuclear translocation, thus leading to an increased expression of oncogenes and inflammatory genes belonging to the Wnt pathway, resulting in a powerful proliferative signal in CRC cells [31,46,47,48,49,50,51]. Such signaling cascades often involve the dysregulation of key oncogenes like *c-MYC*, which has significant implications for therapeutic resistance in gastrointestinal malignancies [52]. The Fap2 protein instead binds to tumor-associated D-galactose-β(1–3)-N-acetyl-D-galactosamine (Gal-GalNAc), a glycan enriched in CRC and other adenocarcinomas, including GC. On the other hand, it enhances immune evasion, engaging the TIGIT inhibitory receptor on natural killer (NK) and T cells [53,54,55].

Importantly, while most mechanistic insights have been derived from CRC, *F. nucleatum* has also been identified in oral, head and neck, esophageal, cervical, and GC tissues [17,39,41,56,57]. This broad oncogenic tropism is partly explained by the affinity of the Fap2 protein to Gal-GalNAc, expressed by many adenocarcinomas. Notably, *F. nucleatum*-positive tumors are associated with higher recurrence rates and poorer prognosis, underscoring its broader oncological relevance and the need for further investigation [44].

## 4. Dysbiosis-Driven Carcinogenesis: Evidence for *F. nucleatum* Involvement in GC

The human gastrointestinal tract hosts a highly diverse and metabolically active microbiota, composed of over 1500 microbial species [1,2]. Disruptions in microbial balance—driven by antibiotics, smoking, diet, or physical inactivity—can result in dysbiosis, a condition linked to inflammatory, metabolic, and neoplastic diseases [7,8,9,58,59,60].

The stomach also harbors its own microbiota, despite having long been considered a hostile environment for microbial proliferation, due to its low luminal pH (1.5–3.5) and high proteolytic enzyme activity. Indeed, a distinct bacterial community persists within the gastric mucus layer, where the pH is relatively more neutral [61,62,63]. External factors, particularly *H. pylori* infection as well as the use of proton pump inhibitors, can raise the gastric pH above 6, thereby promoting microbial overgrowth and altering the gastric microbial landscape [14,64]. In parallel, physiological changes occurring during gastric carcinogenesis—such as inflammation, glandular atrophy, and reduced acid and enzyme secretion—further facilitate microbial colonization [65,66]. These conditions contribute to a shift in the microbial community, culminating in dysbiosis, a state of microbial imbalance increasingly implicated in gastric oncogenesis.

Indeed, several studies have demonstrated that microbial richness and diversity are generally reduced in GC patients, not only in gastric samples but also in oral and fecal specimens, indicating a systemic dysbiotic profile [67,68,69,70]. Interestingly, within tumor tissues, microbial diversity is often higher compared to adjacent non-tumorous areas, suggesting that the tumor microenvironment may selectively foster microbial expansion or harbor specific microbial communities that interact with the host tissue [71,72].

Recent studies have documented major compositional changes in gastric microbiota during GC setting, correlating with disease progression. Increased abundance of genera such as *Fusobacterium*, *Clostridium*, and *Lactobacillus* has been observed in GC tissues [17,72,73], alongside a progressive decline in microbial diversity across histological stages from chronic gastritis to intestinal metaplasia and ultimately to GC [74]. In this context, *F. nucleatum* has emerged as a notable GC-associated taxon, enriched in tumors and also in the gastric juice of patients with advanced GC [17,18,39], and often co-detected with other oral microbes like *Veillonella*, *Leptotrichia*, and *Campylobacter* [11,75].

The role of *F. nucleatum* in gastric carcinogenesis is thought to be exerted in both *H. pylori*-related and *H. pylori*-unrelated GC [16,76]. In the former, *H. pylori*-induced chronic inflammation, glandular atrophy, and increased gastric pH create a more favorable environment for secondary colonizers. This suggests a model of ecological succession rather than direct competition, whereby *H. pylori* facilitates conditions that *F. nucleatum* can later exploit, particularly as the *H. pylori* population declines in advanced cancer stages. On the other hand, an enrichment in *F. nucleatum* is especially notable in *H. pylori*-negative GC, where it seems to be associated with advanced disease, larger tumors, and older age [77,78].

*F. nucleatum* is thought to invade neoplastic gastric cells via endocytosis and to activate the IL−17/NF-κB/RelB signaling pathway, leading to increased IL−17 secretion. The latter promotes the recruitment of N2 TANs, which express PD-L1 and interact with the PD−1 receptor on CD8⁺ T cells, thereby suppressing the anti-tumor immune response and facilitating immune evasion. Concurrently, the bacterium can directly modulate the cytoskeleton, leading to a more invasive cellular phenotype (Figure 2, created in BioRender. Ingravallo, G. (2025) https://BioRender.com/l2px31g, accessed on 1 July 2025).

The pathobiological mechanisms underlying its carcinogenic role remain incompletely understood. However, emerging evidence suggests a direct role of the bacterium in modulating molecular pathways and the immune microenvironmental context that contribute to tumor progression. Mechanistically, as demonstrated in CRC, *F. nucleatum* can invade epithelial cells via endocytosis. Subsequently, lipopolysaccharide (LPS) and other pathogen-associated molecular patterns (PAMPs) engage Toll-like receptor 4 (TLR4) and other pattern recognition receptors, triggering the MyD88-dependent signaling cascade. This culminates in the activation of the NF-κB transcription factor, a master regulator of inflammation, leading to the production of a suite of pro-inflammatory cytokines, including IL−8, IL−18, IL−6, TNF-α, and, notably, IL−17. The latter has been shown to act as a potent chemoattractant for neutrophils which are also within the GC microenvironment, promoting the recruitment and polarization of TANs toward a pro-tumor N2 phenotype, which is characterized by high expression of programmed death-ligand 1 (PD-L1) [18,19,20,79,80,81]. The latter event on one hand promotes immune suppression of anti-tumor CD8+ T cell responses and engages the PD−1 receptor on cytotoxic CD8+ T cells, effectively suppressing the anti-tumor immune response [82,83,84] and facilitating immune evasion and tumor growth, and on the other hand is associated with a better response to the anti-PD-L1 therapy [18,85,86,87]. Transcriptomic analysis of *F. nucleatum*-stimulated neutrophils identified 36 genes overlapping with GC-associated genes involved in protein folding, vesicle trafficking, and endoplasmic reticulum stress (ER stress) pathways [88]. Moreover, in vitro, *F. nucleatum* promotes cytoskeletal changes and increased motility due to the actin regulation, thus leading to an invasive phenotype, as well as promoting inflammatory gene expression [89]. Overall, these results support a broader role in immunosuppression, angiogenesis, and epithelial–mesenchymal transition (EMT), thus promoting a tumor-permissive microenvironment.

Moreover, extracellular vesicles (EVs) represent an additional point of interest, which are small, membrane-bound particles naturally released by almost all cell types. They act as a form of intercellular communication, transporting various molecules like proteins, lipids, and nucleic acids between cells [90]. EVs have emerged as key mediators of *F. nucleatum* pathogenicity by enhancing tumor growth, metastasis, and chemoresistance through the upregulation of stemness and DNA repair markers [91]. The release of EVs from infected host cells represents a potent means of promoting malignancy: EVs enriched with long non-coding RNA HOTTIP increase PI3K-Akt signaling in other non-infected cancer cells, thus enhancing their proliferation, invasion, and migration capacities [19].

The role of EVs extends beyond the transfer of long non-coding RNAs. Recent studies have shown that EVs derived directly from *F. nucleatum* (*Fn*-EVs), not just from infected host cells, are potent drivers of malignant phenotypes. These bacterially derived vesicles are enriched in GC tissues and can be taken up by cancer cells [91,92]. Once internalized, *Fn*-EVs have been shown to enhance chemoresistance to drugs like oxaliplatin, promote cancer cell proliferation and stemness, and increase migratory and invasive capabilities [91,92]. In vivo models have confirmed that administration of *Fn*-EVs alone can promote tumor growth and liver metastasis [91,92]. This indicates that *F. nucleatum* can remodel the tumor microenvironment not only through direct infection but also through the paracrine signaling mediated by its own secreted vesicles, which act as vehicles for delivering virulence factors.

From a clinical perspective, *F. nucleatum* has been linked to TMB, risk of thromboembolic events, and poorer prognosis.

The association between *F. nucleatum* and increased TMB appears to be multifactorial, potentially arising from both indirect and direct effects on genomic integrity.

Indirectly, the chronic inflammation induced by the bacterium contributes to a tumor microenvironment enriched with reactive oxygen species (ROS), such as hydrogen sulfide, which can lead to oxidative DNA damage [89,93].

Furthermore, although not yet demonstrated in GC, studies in other malignancies suggest that *F. nucleatum* may directly induce DNA double-strand breaks, as evidenced by the upregulation of γ-H2AX, a well-established marker of DNA damage, in cancer cell lines [94,95]. Another proposed mechanism involves the suppression of DNA mismatch repair (MMR) pathways: in fact, *F. nucleatum* has been shown to downregulate essential MMR proteins, such as MSH2, thereby contributing to microsatellite instability (MSI)—a key feature associated with high TMB [96,97]. Therefore, by fostering a mutagenic inflammatory environment and potentially interfering with DNA repair machinery, *F. nucleatum* accelerates the accumulation of mutations, contributing to the high TMB phenotype observed in associated tumors [95,97]. In this context, Hsieh et al. [95] reported that co-infection with *H. pylori* and *F. nucleatum* correlates with higher TMB and an increased frequency of mutations in key cancer-related genes, including *ERBB2*, *ERBB3*, *PIK3CA*, and *TP53*. Moreover, the combination of *F. nucleatum* colonization and TMB > 50 mutations/Mb has been linked to unfavorable outcomes, suggesting a synergistic biomarker role for *F. nucleatum* in guiding therapeutic strategies, including immunotherapy.

Liu et al. [78] demonstrated, in a large retrospective cohort study, that *F. nucleatum* colonization was significantly associated with splanchnic vein thrombosis, higher platelet–lymphocyte ratio (PLR), and lower absolute lymphocyte count. The prothrombotic activity of *F. nucleatum* can be ascribed first to induction of neutrophil extracellular traps (NETs) formation, as demonstrated on GC and adjacent tumor tissue by immunohistochemical expression of anti-H3Cit. In turn, neutrophils, activated during NET formation, release the protein 14−3−3ε through EVs, which have been shown to induce megakaryopoiesis via GPIa/PI3K-Akt signaling, thereby increasing platelet production and reinforcing the prothrombotic profile [98].

These biological effects have led to interest in *F. nucleatum* as a biomarker: Boehm et al. [76] firstly found that it is associated with poor prognosis and LINE−1 hypomethylation in diffuse-type GC. More recently, *F. nucleatum* colonization has been identified as an independent prognostic risk factor in GC, based on multivariable Cox regression analysis [78]. Elevated *F. nucleatum* levels in saliva and gastric tissues have also been associated with poor prognosis, particularly in diffuse-type GC, highlighting its non-invasive diagnostic potential [76,99]. Its detection correlates with EMT marker expression and lymph node metastasis [99]. Moreover, microbial signatures combining *F. nucleatum* with other anaerobes (e.g., *Clostridium colicanis*) have demonstrated high sensitivity in GC diagnosis based on gastric biopsies, although specificity remains limited [17].

## 5. Methods for the Detection of *F. nucleatum* in the Stomach: Where We Stand

The detection and quantification of *F. nucleatum* in the gastric environment typically rely on a range of molecular techniques, each contributing to the growing body of evidence regarding its role in gastric oncobiology (Table 1). Gastric tissue specimens, obtained via endoscopic biopsies from both tumors and adjacent non-tumorous mucosa of GC patients, serve as the primary source material. These samples may be processed as either fresh-frozen tissue or formalin-fixed, paraffin-embedded (FFPE) blocks, with the latter being particularly valuable for retrospective analyses. In other cases, non-invasive specimens such as saliva have also been investigated, offering promising potential for biomarker-based screening strategies.

Among the most frequently employed methods, qPCR has proven to be indispensable. By targeting bacterial-specific genes such as the 16S rRNA or *nusG*, qPCR has enabled sensitive and reproducible detection of *F. nucleatum* in both fresh and FFPE tissue samples [39,77,78]. This technique not only facilitates quantification but also allows for correlation between bacterial load and clinical parameters, such as tumor subtype and prognosis. Recently, large-scale epidemiological studies by Kamali et al. [16] and Nascimento Araujo et al. [77] have validated the utility of PCR-based detection in diverse patient populations, providing a broader perspective on prevalence and distribution patterns. In parallel, 16S rRNA gene sequencing has played a crucial role in profiling the gastric microbiome. This culture-independent method targets variable regions (e.g., V3–V4) of the bacterial 16S gene to allow species-level taxonomic classification. Studies like that of Hsieh et al. [17] utilized this method via Illumina MiSeq platforms to detect *F. nucleatum* and co-occurring species such as *Clostridium colicanis*, revealing distinct microbial compositions between tumor and non-tumor regions.

To enhance both sensitivity and specificity—particularly in samples with low bacterial biomass or potential contamination by other microbes—nested PCR protocols have been adopted. For instance, Hsieh et al. [89,95] employed nested PCR targeting of the *nusG* gene to amplify *F. nucleatum* DNA from GC specimens. Subsequent confirmation through gel electrophoresis and sequencing ensured specificity and reliability. Such nested approaches serve as valuable complements when traditional qPCR may yield inconclusive results due to low template concentration. Furthermore, Meng et al. [91] used qPCR to quantify *F. nucleatum* DNA and investigated *Fn*-EVs to assess their impact on chemoresistance.

A more recent advancement, droplet digital PCR (ddPCR), offers absolute quantification of bacterial DNA with high sensitivity and reproducibility. Chen et al. [99] applied ddPCR to analyze salivary *F. nucleatum* levels in GC patients, finding elevated concentrations in individuals with advanced disease. However, given that *F. nucleatum* is a common constituent of the oral microbiota, the diagnostic specificity of salivary detection requires further validation to distinguish GC-associated colonization from periodontal disease or healthy carriage. Although this study focused on saliva, the technique itself holds substantial potential for tissue-based analyses, especially where precise quantification is essential.

Beyond DNA-based techniques, FISH has provided critical spatial resolution. This method employs fluorescently labeled oligonucleotide probes targeting *F. nucleatum* DNA to visualize its localization within tissue architecture and allows researchers not only to confirm bacterial presence but also to infer potential host–microbe interactions at the cellular level. Interestingly, it has been applied in combination with qPCR [19,78,98] to confirm the presence of *F. nucleatum* in gastric tumors, often within superficial epithelial layers. Moreover, this integrated platform has been employed to demonstrate the oncogenic potential of *F. nucleatum* via exosomal RNA (HOTTIP) and downstream PI3K/AKT signaling [19].

In terms of functional validation, proteomic approaches such as mass spectrometry have emerged as complementary tools. Aziz et al. [100] employed mass spectrometry to detect *F. nucleatum*-derived proteins in GC tissues, thereby establishing bacterial viability and metabolic activity. This method surpasses mere DNA detection by confirming the transcriptional and translational activity of the pathogen within the tumor microenvironment.

Collectively, these methods—qPCR, qRT-PCR, nested PCR, ddPCR, FISH, proteomics, and 16S rRNA sequencing—offer a multi-layered toolkit for detecting *F. nucleatum* in GC. Each contributes distinct strengths: qPCR and ddPCR for quantification, FISH for spatial localization, proteomics for assessing bacterial activity, and sequencing for extensive microbiota profiling. The integration of these techniques provides a comprehensive framework for molecular microbiology in gastric oncology, paving the way for refined diagnostic and research applications.

## 6. Future Directions

The growing body of evidence implicating *F. nucleatum* in gastric carcinogenesis opens up exciting new avenues for diagnosis, prognosis, and therapy. While its role as a biomarker is promising, the ultimate goal is to translate these findings into clinical interventions that can improve patient outcomes. Several therapeutic strategies targeting the bacterium or its pathogenic mechanisms can be envisioned.

### 6.1. Direct Antimicrobial Strategies

A straightforward approach is the direct eradication of *F. nucleatum* using antibiotics. Metronidazole, an antibiotic effective against anaerobic bacteria, has been shown to reduce Fusobacterium load and attenuate its pro-tumorigenic effects in preclinical colorectal cancer models [96]. This strategy could be particularly relevant for *F. nucleatum*-positive GC patients to reduce tumor growth, prevent recurrence, or even resensitize chemoresistant tumors. However, this approach faces significant challenges, including the risk of promoting antibiotic resistance and the potential for disrupting the beneficial components of the gut microbiota, which could have unintended negative consequences [96,101]. Therefore, the development of more specific, non-antibiotic antimicrobial approaches, such as bacteriophages or targeted antimicrobial peptides, represents a more sophisticated future direction [101].

### 6.2. Targeting Bacterial Virulence Factors

A more refined strategy involves neutralizing the specific virulence factors that mediate *F. nucleatum*’s oncogenic effects. The adhesins FadA and Fap2 are prime candidates. Developing small-molecule inhibitors or monoclonal antibodies that block the interaction of FadA with E-cadherin or Fap2 with its receptors (Gal-GalNAc on tumor cells or TIGIT on immune cells) could disrupt bacterial colonization and dismantle its immunosuppressive shield without killing the bacterium, thus posing a lower risk of inducing resistance [102]. Such targeted therapies could sever the critical link between the bacterium and the host cancer cells, neutralizing its ability to drive proliferation and immune evasion.

### 6.3. Modulating the Host Immune Response and Combination Therapies

Given that *F. nucleatum* profoundly remodels the tumor immune microenvironment, often by inducing PD-L1 expression, combining anti-*F. nucleatum* strategies with immunotherapy is a highly promising avenue [103,104]. The role of *F. nucleatum* in the context of immunotherapy is complex and appears to be context-dependent. While some studies show that its presence is associated with resistance to anti-PD−1/PD-L1 therapy, others in colorectal cancer suggest it can enhance immunotherapeutic efficacy by increasing T cell infiltration [105]. For GC, where *F. nucleatum* promotes an immunosuppressive phenotype, eliminating the bacterium or blocking its pathways could “re-awaken” the anti-tumor immune response. This could convert an immunologically “cold” tumor into a “hot” one, making it more susceptible to immune checkpoint inhibitors [96]. Clinical trials exploring the combination of targeted antibiotics and immunotherapy in biomarker-stratified patient populations are a logical and compelling next step.

### 6.4. Microbiota-Based Interventions

Finally, broader interventions aimed at remodeling the entire gastric microbiota may hold therapeutic potential. Probiotics or fecal microbiota transplantation (FMT) could be explored to restore a healthy microbial equilibrium and competitively exclude or inhibit the growth of *F. nucleatum* and other pathobionts. While still highly speculative for GC, this approach aligns with a growing appreciation for the microbiome as a holistic therapeutic target [96,101]. Further research is needed to identify specific probiotic strains that can effectively antagonize *F. nucleatum* in the gastric niche. In summary, moving forward requires a multi-pronged approach: refining non-invasive diagnostic tools; validating *F. nucleatum*’s prognostic value in large, diverse cohorts; and, most importantly, advancing these potential therapeutic strategies from preclinical models into well-designed clinical trials.

## 7. Conclusions

In conclusion, *F. nucleatum* represents a multifaceted microbial agent in gastric carcinogenesis. Beyond its biological impact on immune modulation, immune evasion, proliferation, invasiveness, and migration, its presence holds promise as a biomarker for diagnosis, prognosis, and potentially therapeutic targeting. However, it must still be considered that inconsistencies across populations and methodologies—particularly between studies from Asia and Western countries—highlight the need for standardization in microbial detection protocols and further validation in multicenter cohorts [68,76,77]. For instance, variations in the prevalence of *F. nucleatum* and its association with specific GC subtypes across different geographic regions currently limit the global applicability of these findings. In this regard, although the great majority of studies have reported an enrichment of *F. nucleatum* in GC tissues, conflicting results persist. Nascimento et al. [77] and Ferreira et al. [68] did not observe significant differences in *F. nucleatum* levels between GC patients and controls, with the latter reporting even lower proportions in GC cases (0.5%) compared to gastritis (1.8%). Notably, both the studies were conducted in non-Asian populations, suggesting that geographic, ethnic, or methodological variability may influence microbial findings. Crucially, the current understanding is limited to gastric adenocarcinoma, and the role of the microbiota in rarer gastric neoplasms remains unknown [106,107,108,109]. Lastly, further research should prioritize translational approaches to validate its clinical utility and clarify its interactions with host immunity and other microbial species in the gastric microenvironment.

## Figures and Tables

**Figure 1 ijms-26-07915-f001:**
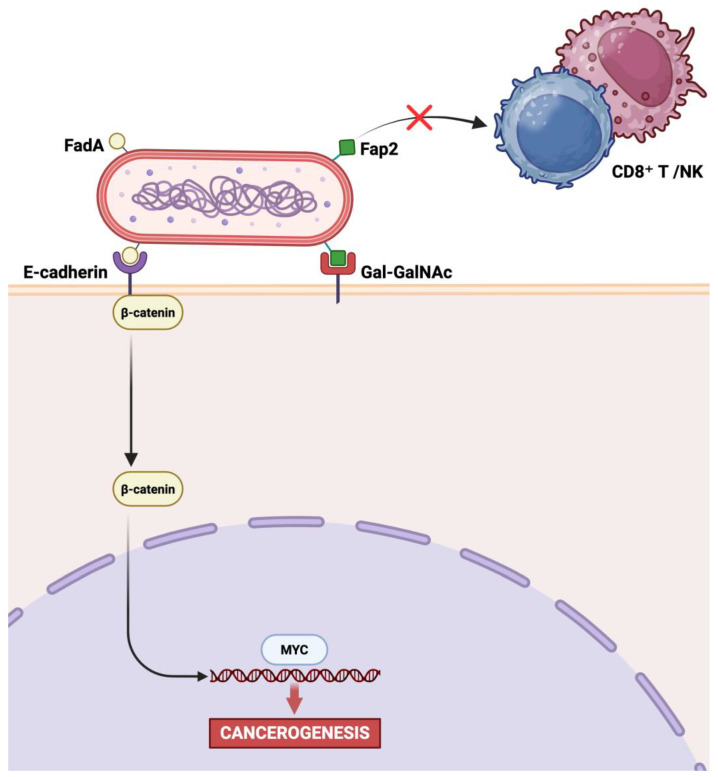
Key virulence factors of *F. nucleatum* and their roles in carcinogenesis. Created in BioRender. Ingravallo, G. (2025) https://BioRender.com/li3ppbl, accessed on 1 July 2025.

**Figure 2 ijms-26-07915-f002:**
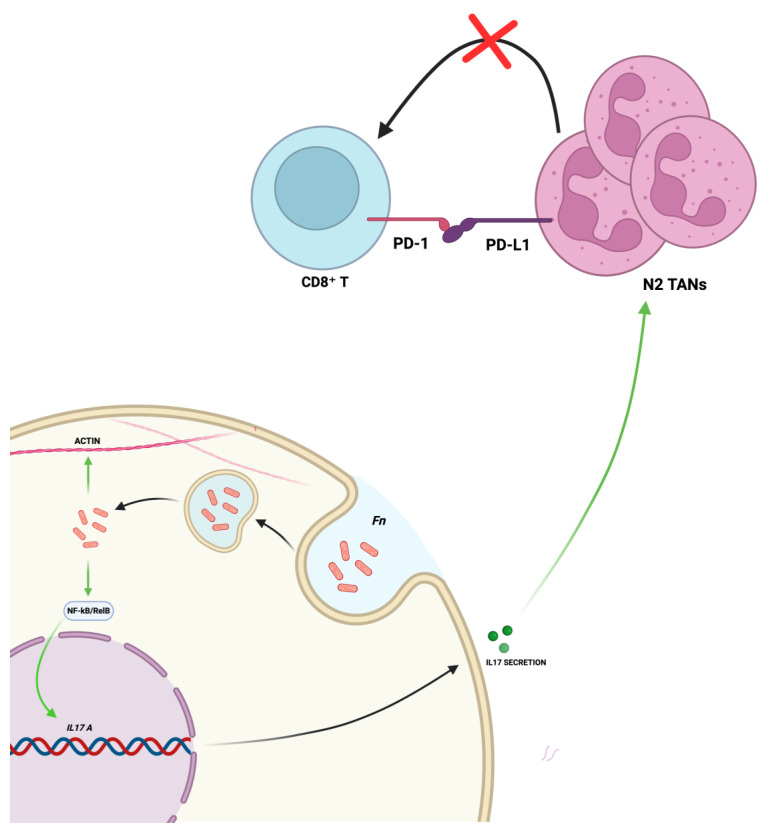
Main effects of *F. nucleatum* on GC cells. Created in BioRender. Ingravallo, G. (2025) https://BioRender.com/l2px31g, accessed on 1 July 2025.

**Table 1 ijms-26-07915-t001:** Comparison of molecular methods for the detection of *Fusobacterium nucleatum* in biological samples.

Method	Target Gene(s)/Marker(s)	Sample Type(s)	Key Strengths
qPCR/qRT-PCR [16,17,39,77,91,98]	*16S rRNA*, *nusG*	Gastric biopsy (fresh or FFPE)	Sensitive, quantitative, high throughput
Nested PCR [89,95]	*nusG*	Gastric biopsy (FFPE)	Enhanced specificity and sensitivity; ideal for low biomass or degraded samples
ddPCR [99]	*nusG*	Saliva	Ultra-sensitive and highly specific; absolute quantification without standard curve
FISH [19,78]	*Fn*-specific DNA probes (5′-CGCAATACAGAGTTGAGCCCTGC−3′) (5′-CTTGTAGTTCCGC(C/T)TACCTC−3′)	Gastric biopsy (FFPE)	Enables spatial localization and visualization of microbial–host interactions
High-definition mass spectrometry	*Fn*-specific proteins (atpD, FN0857, FN1974, FN0813, clpB, FN1649, FN1546)	Gastric biopsy (FFPE)	Confirms viability and metabolic activity of bacteria; proteomic-level specificity

This table summarizes the main molecular approaches used to detect *F. nucleatum*, detailing the target genes, sample types, and key strengths of each method. Techniques include DNA or RNA amplification assays (qPCR, qRT-PCR, nested PCR, ddPCR), in situ hybridization (FISH), and high-definition mass spectrometry for protein detection. The methods are directed toward gene sequences or proteins that are highly specific to *F. nucleatum*.

## Data Availability

No new data were created or analyzed in this study.

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
