# Peer review of "Fusobacterium nucleatum and Gastric Cancer: An Emerging Connection"

_ijms, 2025, doi:10.3390/ijms26167915_

Round 1

Reviewer 1 Report

Comments and Suggestions for Authors

The article requires serious corrections:
1. Unfortunately, the title is misleading, as the role of Fusobacterium nucleatum in carcinogenesis has been known for many years, so "New Microbial Player" should be removed. Most of the information regarding the molecular aspects of F. nucleatum's action can be found in the article https://www.sciencedirect.com/science/article/abs/pii/S1044579X22000049.
2. The figures are not fully understandable; they appear copied and therefore lack detail. For example, does Figure 1 indicate that the combination of Fap2 and TIGIT leads to tumor inhibition? Is that the symbol shown in the figure? And today we know that a complex reaction occurs, including inhibition of the immune response along with increased levels of proinflammatory cytokines. This is missing from the figures.
3. The article and figures lack a description of the LPS-TLR4 or ANGPTL4 link. This work lacks new information because the authors should have reviewed the latest data and presented it in the article, which they did not.
4. In the Materials and Methods, the number of keywords used is extremely small. Missing keywords include virulence factors, dysbiosis, tumorigenesis, Helicobacter pylori-independent gastric cancer, tumor microenvironment, bacterial translocation, etc. Therefore, the work is incomplete.
5. In particular, the text in chapter "4. Dysbiosis-Driven Carcinogenesis" needs to be improved because the links between individual paragraphs are missing.
6. In chapter 5, a description of how material for molecular testing is collected from the stomach is missing. It would also be worthwhile to include a table specifying the type of test, primers used, reaction conditions, etc.

Comments on the Quality of English Language

English should be corrected by native speaker.

Author Response

Cover Letter for Resubmission

Dear Editor,

We thank you and the reviewers for the insightful comments on our manuscript entitled: "The Emerging Role of Fusobacterium nucleatum in Gastric Carcinogenesis: A New Microbial Player in the Tumor Microenvironment". We believe the feedback has been invaluable in improving the quality, clarity, and completeness of our review.

We have addressed all the points raised by the reviewers and have revised the manuscript accordingly. A point-by-point response to the reviewers' comments is provided below, detailing the changes made. The revised sections of the manuscript are also included for your convenience.

We believe that the revised manuscript is now significantly strengthened and hope that it is suitable for publication in the International Journal of Molecular Sciences.

Sincerely,

The Authors

Point-by-Point Response to Reviewer 1

We thank the reviewer for the thorough evaluation and constructive feedback. Below we address each comment in detail.

Response to Reviewer 1

Reviewer 1, Comment 1:
"Unfortunately, the title is misleading, as the role of Fusobacterium nucleatum in carcinogenesis has been known for many years, so 'New Microbial Player' should be removed. Most of the information regarding the molecular aspects of F. nucleatum's action can be found in the article https://www.sciencedirect.com/science/article/abs/pii/S1044579X22000049."

Our Response:
We agree with the reviewer. While the role of F. nucleatum in gastric cancer is more "emerging" compared to its well-established role in colorectal cancer, the term "new" could be misleading. We have revised the title to be more precise.

  • Original Title: The Emerging Role of Fusobacterium nucleatum in Gastric Carcinogenesis: A New Microbial Player in the Tumor Microenvironment
  • Revised Title: Fusobacterium nucleatum and Gastric Cancer: An Emerging Connection

Reviewer 1, Comment 2:
"The figures are not fully understandable; they appear copied and therefore lack detail. For example, does Figure 1 indicate that the combination of Fap2 and TIGIT leads to tumor inhibition? Is that the symbol shown in the figure? And today we know that a complex reaction occurs, including inhibition of the immune response along with increased levels of proinflammatory cytokines. This is missing from the figures."

Our Response:
We thank the reviewer for pointing out the ambiguity in Figures. We have revised both the figures, to clarify molecular mechanism and have expanded the corresponding text to include the complexity of the immune response.

Reviewer 1, Comment 3:
"The article and figures lack a description of the LPS-TLR4 or ANGPTL4 link. This work lacks new information because the authors should have reviewed the latest data and presented it in the article, which they did not."

Our Response:
We appreciate this comment. Our manuscript already described the engagement of Toll-like receptor 4 (TLR4) in Section 6, which leads to NF-κB activation. However, we acknowledge that the role of Lipopolysaccharide (LPS) and the potential link to ANGPTL4 were not explicitly detailed. We have performed a literature search on the F. nucleatum-ANGPTL4 link in gastric cancer. While this link is emerging in other contexts, it is not yet well-established for F. nucleatum in gastric cancer specifically. To maintain the focus and accuracy of the review, we have expanded the discussion on the LPS-TLR4 axis but have refrained from speculating on the ANGPTL4 link until more direct evidence becomes available.

  • Revised Text (lines 212-218):
    "Mechanistically, F. nucleatum can invade epithelial cells via endocytosis. Upon invading gastric epithelial cells, its Lipopolysaccharide (LPS) and other pathogen-associated molecular patterns (PAMPs) engage Toll-like receptor 4 (TLR4) and other pattern recognition receptors, triggering the MyD88-dependent signaling cascade. This culminates in the activation of the NF-κB transcription factor, a master regulator of inflammation, leading to the production of a suite of pro-inflammatory cytokines, including IL-8, TNF-α, and importantly, IL-17."

Reviewer 1, Comment 4:
"In the Materials and Methods, the number of keywords used is extremely small. Missing keywords include virulence factors, dysbiosis, tumorigenesis, Helicobacter pylori-independent gastric cancer, tumor microenvironment, bacterial translocation, etc. Therefore, the work is incomplete."

Our Response:
We agree and have expanded the list of keywords to better reflect the scope of the manuscript.

  • Original Keywords: Fusobacterium nucleatum; gastric cancer; gastric microbiota.
  • Revised Keywords:Fusobacterium nucleatum” “gastric cancer,” “gastric microbiota”, “gastric dysbiosis”, “gastric tumorigenesis”, “gastric tumor microenvironment” and “Helicobacter pylori-independent gastric cancer”.

Reviewer 1, Comment 5:
"In particular, the text in chapter '4. Dysbiosis-Driven Carcinogenesis' needs to be improved because the links between individual paragraphs are missing."

Our Response:
Thank you for this valuable suggestion. We have revised Chapter 4 to improve the narrative flow and create clearer transitions between paragraphs, ensuring a more cohesive argument.

  1. Modified text (lines 166-169):

"The stomach also harbors its own microbiota, despite having long been considered a hostile environment for microbial proliferation, due to its low luminal pH (1.5–3.5) and high proteolytic enzyme activity. Indeed, a distinct bacterial community persists within the gastric mucus layer, where the pH is relatively more neutral [61–63].”

  1. Modified text (lines 196-198):

“On the other hand, an enrichment in F. nucleatum is especially notable in H. pylori-negative GC, where seems to be associated with advanced disease, larger tumors, and older age [77,78].”

  1. Modified text (lines 209-221):

“The pathobiological mechanisms underlying its carcinogenic role remain incompletely understood. However, emerging evidence suggests a direct role for the bacterium in modulating molecular pathways and the immune–microenvironmental context that contribute to tumor progression. Mechanistically, F. nucleatum can invade epithelial cells via endocytosis. Upon invading gastric epithelial cells, its Lipopolysaccharide (LPS) and other pathogen-associated molecular patterns (PAMPs) engage Toll-like receptor 4 (TLR4) and other pattern recognition receptors, triggering the MyD88-dependent signaling cascade. This culminates in the activation of the NF-κB transcription factor, a master regulator of inflammation, leading to the production of a suite of pro-inflammatory cytokines, including IL-8, TNF-α, and importantly, IL-17 [79,80]. The secreted IL-17 acts as a potent chemoattractant for neutrophils, leading to the recruitment and polarization of TANs [81], that adopt a pro-tumor N2 phenotype and express high levels of Programmed Death-Ligand 1 (PD-L1).”

  1. Modified text (lines 253-274):

“From a clinical perspective, F. nucleatum has been linked to TMB, risk of thromboembolic events, and poorer prognosis.

The association between F. nucleatum and increased TMB appears to be multifactorial, potentially arising from both indirect and direct effects on genomic integrity.

Indirectly, the chronic inflammation induced by the bacterium contributes to a tumor microenvironment enriched with reactive oxygen species (ROS), such as hydrogen sulfide, which can lead to oxidative DNA damage [89, 93].

Furthermore, although not yet demonstrated in GC, studies in other malignancies suggest that F. nucleatum may directly induce DNA double-strand breaks, as evidenced by the upregulation of γ-H2AX, a well-established marker of DNA damage, in cancer cell lines [94,95]. Another proposed mechanism involves the suppression of DNA mismatch repair (MMR) pathways: in fact, F. nucleatum has been shown to downregulate essential MMR proteins, such as MSH2, thereby contributing to microsatellite instability (MSI)—a key feature associated with high TMB [96, 97]. Therefore, by fostering a mutagenic inflammatory environment and potentially interfering with DNA repair machinery, F. nucleatum accelerates the accumulation of mutations, contributing to the high TMB phenotype observed in associated tumors [95, 97]. In this context, Hsieh et al. [95] reported that co-infection with H. pylori and F. nucleatum correlates with higher TMB and an increased frequency of mutations in key cancer-related genes, including ERBB2, ERBB3, PIK3CA, and TP53. Moreover, the combination of F. nucleatum colonization and TMB > 50 mutations/Mb has been linked to unfavorable outcomes, suggesting a synergistic biomarker role for F. nucleatum in guiding therapeutic strategies, including immunotherapy.”

Reviewer 1, Comment 6:
"In chapter 5, a description of how material for molecular testing is collected from the stomach is missing. It would also be worthwhile to include a table specifying the type of test, primers used, reaction conditions, etc."

Our Response:
This is an excellent suggestion. We have added a paragraph at the beginning of Chapter 5 to describe sample collection and have created a new table (Table 1) to summarize the key molecular detection methods as requested.

  • Added Text (lines 296-304):
    “The detection and quantification of nucleatum in the gastric environment typically rely on a range of molecular techniques, each contributing to the growing body of evi-dence regarding its role in gastric oncobiology. Gastric tissue specimens, obtained via endoscopic biopsies from both tumor and adjacent non-tumorous mucosa of GC patients, serve as the primary source material. These samples may be processed as either fresh-frozen tissue or formalin-fixed, paraffin-embedded (FFPE) blocks, with the latter being particularly valuable for retrospective analyses. In addition to tissue samples, non-invasive specimens such as saliva and gastric juice have also been investigated, offering promising potential for biomarker-based screening strategies.”
  • New Table Added (see Table 1)

Reviewer 1, Comment on Language:
"Comments on the Quality of English Language: English should be corrected by native speaker."

Our Response:
We thank the reviewer for this comment. The entire manuscript has been carefully reviewed and edited for English language, grammar, and clarity by a native English speaker to ensure it meets the publication standards of the journal.

We are confident that these revisions have substantially improved the manuscript. We look forward to your positive evaluation.

Reviewer 2 Report

Comments and Suggestions for Authors

Dear Editor,

I read with great interest the manuscript entitled: The Emerging Role of Fusobacterium nucleatum in Gastric Carcinogenesis: A New Microbial Player in the Tumor Microenvironment.
The review is valuable and addresses an increasingly important topic in the field of oncology, summarizing data which is fundamental in understanding the pathophysiology of gastric cancer and possible ways to prevent, detect or even treat it.

Following are some comments to make the manuscript more appealing:

  1. I believe the Manuscript would benefit substantially of a more detailed description of the clinical studies in which Nucleatum was detected. Specifically, data should be added on ways in which it was detected (e.g. mucosal biopsy vs gastric aspirate)
  2. Please add information on different stages in which stages of gastric cancer was detected.
  3. Please mention whether this bacterium is present in several diagnostic models that have been elaborated for the non-invasive detection of gastric cancer.
  4. The Authors performed structured research to find relevant manuscripts for the review. It could be useful to explicit the number of studies considered.
  5. Mention studies that failed to detect the bacterium in gastric tissues or found no association with prognosis.
  6. Discuss the influence of oral health and periodontal disease, both in Nucleatum detection and its possible therapeutic role.
  7. Minor comment: chapter 4 introduction is redundant, please reduce the repetitions.

Author Response

Cover Letter for Resubmission

Dear Editor,

We thank you and the reviewers for the insightful comments on our manuscript entitled: "The Emerging Role of Fusobacterium nucleatum in Gastric Carcinogenesis: A New Microbial Player in the Tumor Microenvironment". We believe the feedback has been invaluable in improving the quality, clarity, and completeness of our review.

We have addressed all the points raised by the reviewers and have revised the manuscript accordingly. A point-by-point response to the reviewers' comments is provided below, detailing the changes made. The revised sections of the manuscript are also included for your convenience.

We believe that the revised manuscript is now significantly strengthened and hope that it is suitable for publication in the International Journal of Molecular Sciences.

Sincerely,

The Authors

Point-by-Point Response to Reviewer 2

We thank the reviewer for the thorough evaluation and constructive feedback. Below we address each comment in detail.

Reviewer 2, Comment 1:

"I believe the Manuscript would benefit substantially of a more detailed description of the clinical studies in which Nucleatum was detected. Specifically, data should be added on ways in which it was detected (e.g. mucosal biopsy vs gastric aspirate)"

Our Response:

We agree with this assessment. As addressed in our response to Reviewer 1 (Comment 6), we have added a new paragraph at the start of Chapter 5 describing the types of clinical samples used for detection (mucosal biopsies, FFPE tissues, saliva) and have included this information in the new summary table (Table 1).

Reviewer 2, Comment 2:

"Please add information on different stages in which stages of gastric cancer was detected."

Our Response:

Thank you for this suggestion. We have revised the manuscript to be more specific about the association of F. nucleatum with advanced stages of gastric cancer.

  • Revised Text (Abstract, line 17):

"...particularly in patients negative for Helicobacter pylori (H. pylori) or in advanced-stage GC cases, F. nucleatum exerts diverse oncogenic effects."

  • Revised Text (Section 4, last paragraph, line 188-189):

"...In this context, F. nucleatum has emerged as a notable GC-associated taxon, enriched in tumors and also in gastric juice of patients with advanced GC, particularly those with larger tumors and at a later clinical stage..."

Reviewer 2, Comment 3:

"Please mention whether this bacterium is present in several diagnostic models that have been elaborated for the non-invasive detection of gastric cancer."

Our Response:

This is an important point. We have expanded the discussion on non-invasive detection, specifically highlighting the use of saliva as a diagnostic fluid and citing relevant studies.

  • Revised Text (lines 328-336):

“A more recent advancement, droplet digital PCR (ddPCR), offers absolute quantification of bacterial DNA with high sensitivity and reproducibility. Chen et al. [81] applied ddPCR to analyze salivary F. nucleatum levels in GC patients, finding elevated concentrations in individuals with advanced disease. However, given that F. nucleatum is a common constituent of the oral microbiota, the diagnostic specificity of salivary detection requires further validation to distinguish GC-associated colonization from periodontal disease or healthy carriage. Although this study focused on saliva, the technique itself holds substantial potential for tissue-based analyses, especially where precise quantification is essential.”

Reviewer 2, Comment 4:

"The Authors performed structured research to find relevant manuscripts for the review. It could be useful to explicit the number of studies considered."

Our Response:

We appreciate this suggestion for improving the methods section. We have revised the text to provide a clearer scope of our literature search.

Revised Text (Section 2. Materials and Methods):

“A research was carried out on public scientific databases (PubMed, Scopus and Web of Science) up to May 2025, employing the following keywords: “Fusobacterium nucleatum,” “gastric cancer,” “gastric microbiota”, “gastric dysbiosis”, “gastric tumorigenesis”, “gastric tumor microenvironment” and “Helicobacter pylori-independent gastric cancer”. Only articles published in English were included. Studies were screened for relevance based on titles, abstracts, and full texts. Priority was given to original research articles with experimental, clinical, or metagenomic evidence on the presence or role of F. nucleatum in gastric tumorigenesis.”

Reviewer 2, Comment 5:

"Mention studies that failed to detect the bacterium in gastric tissues or found no association with prognosis."

Our Response:

This is a critical point for a balanced review. We have added a statement in the Conclusion to acknowledge the inconsistencies in the literature and the need for further research.

  • Added Text (lines 432-438):

“On this regard, although the great majority of studies have reported an enrichment of F. nucleatum in GC tissues, conflicting results persist. Nascimento et al. [77] and Ferreira et al. [68] did not observe significant differences in F. nucleatum levels between GC patients and controls, with the latter reporting even lower proportions in GC cases (0.5%) compared to gastritis (1.8%). Notably, both the studies were conducted in non-Asian populations, suggesting that geographic, ethnic, or methodological variability may influence microbial findings.”

Reviewer 2, Comment 6:

"Discuss the influence of oral health and periodontal disease, both in Nucleatum detection and its possible therapeutic role."

Our Response:

Dear Reviewer, we have already briefly discussed the role of Fusobacterium nucleatum in oral biofilm formation and periodontitis (Chapter 3). However, we have intentionally chosen not to focus on its involvement in this context or in other pathological conditions, as our objective is to concentrate specifically on its role in gastric cancer. The association between F. nucleatum and periodontal diseases has already been extensively addressed in dedicated studies throughout the literature.

Reviewer 2, Comment 7:

"Minor comment: chapter 4 introduction is redundant, please reduce the repetitions."

Our Response:

We agree with the reviewer's observation. The introduction to Chapter 4 repeated some information from the main introduction. We have condensed it to be more direct and to serve as a better entry point to the section's specific focus.

  • Original Text (Section 4, first paragraph):

"The human gastrointestinal tract hosts a highly diverse and metabolically active microbiota, composed of over 1,500 microbial species. Dominated by Bacteroidetes and Firmicutes, along with other phyla (e.g., Proteobacteria, Fusobacteria), this ecosystem supports host health through vitamin production, immune modulation, and epithelial barrier maintenance [3-5]. Disruptions in microbial balance—driven by antibiotics, smoking, diet, or physical inactivity—can result in dysbiosis, a condition linked to inflammatory, metabolic, and neoplastic diseases [7–9,58–60]."

  • Revised Text (Section 4, first paragraph, lines 162-165):

The human gastrointestinal tract hosts a highly diverse and metabolically active microbiota, composed of over 1,500 microbial species [1,2]. Disruptions in microbial balance—driven by antibiotics, smoking, diet, or physical inactivity—can result in dysbiosis, a condition linked to inflammatory, metabolic, and neoplastic diseases [7–9,58–60].

We are confident that these revisions have substantially improved the manuscript. We look forward to your positive evaluation.

Round 2

Reviewer 1 Report

Comments and Suggestions for Authors

The authors have improved their manuscript, which I am very pleased with. They have taken most of my comments into account.

1. I suggest that the authors add a few more sentences in Chapter 4, regarding the effect of F. nucleatum on the activation of proinflammatory cytokines (IL-1b, IL-6, TNF-a) and the development of chronic inflammation. This data could also be added to the figure. Please see Fig. 5 in the attached article https://doi.org/10.1016/j.semcancer.2022.01.004.

2. Please also correct the name "F. Nucleatum" to "F. nucelatum" in some places – the species name is lowercase.

Author Response

Dear Reviewer,

We have revised the section of Chapter 4 dedicated to cytokines to make it clearer. The passage now serves as a bridge to explain, based on the most consolidated findings in CRC, the production of IL-17. We have also reformulated the text in lines 307–318 to avoid potential ambiguities.

As per our initial intention, we did not add further graphical information to the figure, since some mechanisms (as specified in the text) have been demonstrated in CRC but not yet fully in the stomach — as accurately represented in the image.

Moreover, we have included the citation of the article you kindly suggested, to complete the explanation in the aforementioned lines.

Finally, we have corrected “F. Nucleatum” to “F. nucleatum” throughout the text.